**Brief Communication**

# SyConn2: dense synaptic connectivity inference for volume electron microscopy

Philipp J. Schubert [1], Sven Dorkenwald[1,2,3], Michał Januszewski [4], Jonathan Klimesch[1], Fabian Svara[5,8], Andrei Mancu [1], Hashir Ahmad [1], Michale S. Fee [6], Viren Jain [7] and Joergen Kornfeld [1] ✉

The ability to acquire ever larger datasets of brain tissue using volume electron microscopy leads to an increasing demand for the automated extraction of connectomic information. We introduce SyConn2, an open-source connectome analysis toolkit, which works with both on-site high-performance compute environments and rentable cloud computing clusters. SyConn2 was tested on connectomic datasets with more than 10 million synapses, provides a web-based visualization interface and makes these data amenable to complex anatomical and neuronal connectivity queries.

The acquisition speed of state-of-the-art volume electron microscopy (VEM) has increased about 100-fold during the past five years[1], and petabyte-scale datasets have been generated[2,3]. The associated computational challenges can only be addressed with scalable analysis frameworks, requiring either classical high-performance computing environments or commercial compute cloud offerings, and are facilitated by open-source code that also increases reproducibility.

Despite these increases in acquisition speed and considerable advances in areas such as automated neuron reconstruction[4], proofreading[5], synapse and organelle detection[6,7], cell type classification[8,9] and integrative processing in cloud environments[10,11], a pipeline that creates an annotated connectome and can also be operated cost-efficiently on existing high-performance computing infrastructure is lacking. Here we introduce SyConn2, which requires existing dense neuron reconstructions and fundamentally upgrades our earlier software package[7] (see Supplementary Table 1 for a comparison) to allow neuroscientists to run queries against connectomes with millions of synapses[12,13]. To be able to handle the large amounts of data at reasonable cost, we focused on computationally efficient processing at every step, for example by operating on lightweight point cloud representations instead of dense data structures to analyze neuron morphology. The SyConn2 processing speed was about 34 megavoxels per hour per central processing unit (CPU) core and 4.4 gigavoxels per hour per graphics processing unit (GPU). This leads to an approximate cost (evaluated on the Google Cloud Platform using a zebra finch dataset) of about US$2,000 per teravoxel of 8-bit raw VEM data at a voxel size of $10 \times 10 \times 25$ nm$^3$ (approximately US$800 per million μm$^3$).

Taking advantage of the details visible in dense heavy-metal stains of tissue, SyConn2 processing starts with multiple semantic voxel-level annotations spanning the entire VEM dataset, including a segmentation into cells, extracellular space, synaptic locations and organelles, among others (Fig. 1). SyConn2 provides the option to apply deep neural network segmentation models to an entire raw three-dimensional (3D) image dataset for the classes of interest, by splitting it into chunks that are processed in parallel by many CPU and GPU workers, using the SLURM workload manager. To map synaptic connectivity between neurons, contact sites between different cell segmentations are detected (Fig. 1, top), which are then overlapped with the neural network-generated synaptic junction segmentations to form candidate synapse objects. Although the overlap of a contact site between two neurons and a detected synaptic cleft could be used, in theory, by itself to decide whether a contact is synaptic, we found it helpful to add further quantitative ultrastructural information from surrounding voxels. These synaptic features can either be used directly for analyses or used for another classification stage (for example, with a random forest classifier[7]) at each candidate synaptic location.

[1]Max Planck Institute of Neurobiology, Martinsried, Germany. [2]Princeton Neuroscience Institute, Princeton, NJ, USA. [3]Computer Science Department, Princeton, NJ, USA. [4]Google Research, Zurich, Switzerland. [5]Max Planck Institute for Neurobiology of Behavior - caesar, Bonn, Germany. [6] Department of Brain and Cognitive Sciences, McGovern Institute for Brain Research, Massachusetts Institute of Technology, Cambridge, MA, USA. [7]Google Research, Mountain View, CA, USA. [8]Present address: ariadne.ai ag, Buchrain, Switzerland. ✉e-mail: joergen.kornfeld@bi.mpg.de

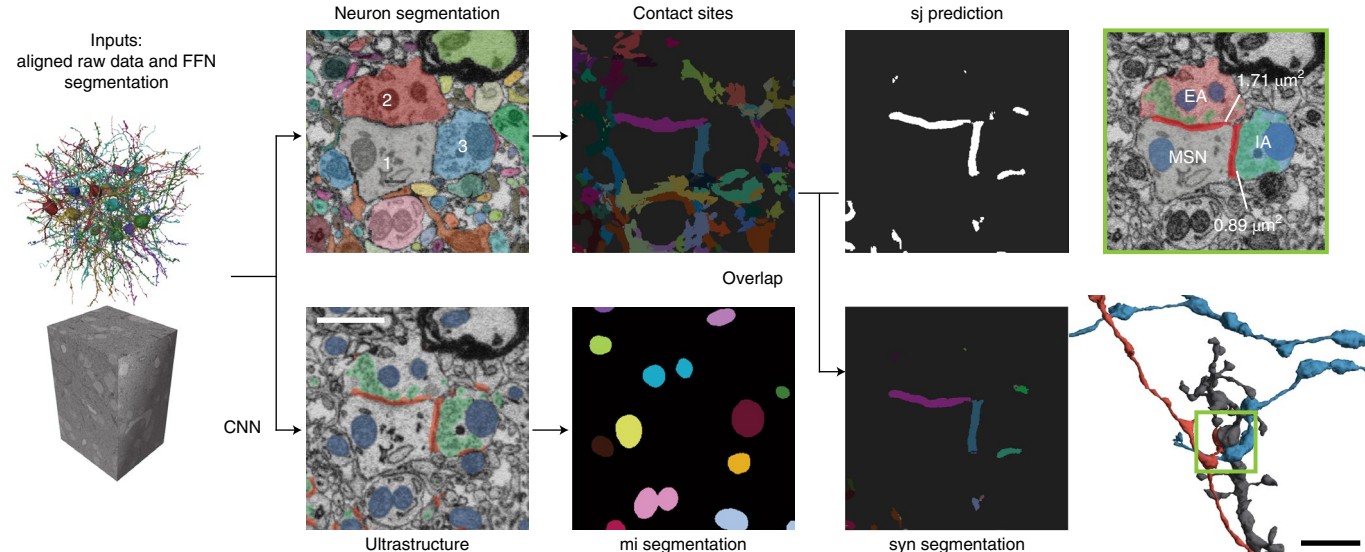

**Fig. 1 | SyConn2 processing on the voxel level.** Neuron segmentation and ultrastructure prediction (synaptic junctions (sj) in red; mitochondria (mi) in blue; vesicle clouds in green) derived from raw data. Contact sites and synaptic junction are assigned as queryable instances to neuron reconstructions. 1, MSN dendrite; 2, excitatory axon (EA); 3, inhibitory axon (IA). The scale bar is 1 μm and refers to all electron microscopy images in the figure. CNN: convolutional neural network ; FFN: flood-filling neural network.

We previously introduced cellular morphology neural networks (CMN) for morphology and type classification based on learning 3D shape from two-dimensional projection images of a neuron[14]. Although the CMN approach provides high accuracy, it suffers from low processing speeds, which becomes increasingly important with growing datasets. To increase processing speed, we use a representation in which cell membranes are represented by sparse discrete points (Fig. 2a). Deep convolutional neural networks trained directly on sparse point representations are well suited; thus, we used the ConvPoint architecture[15] as the basis for our experiments. The resulting morphology classifier performed at the same accuracy level as our previous CMN architecture[14] (Fig. 2b,c, Supplementary Texts 1 and 2 and Supplementary Table 2; see also ref. [16] for a dense 3D approach) with a 3.3-fold higher throughput (Extended Data Fig. 1a). We conducted a detailed throughput and scalability analysis of the entire pipeline (Extended Data Fig. 1 and Supplementary Text 3).

We next explored whether the point-based morphology neural network could also be used for unsupervised (without requiring handcrafted training data) cell type discovery[17] through dimensionality reduction of a learned latent space. We extended our previous approach of triplet-loss morphology learning[14] to generate embeddings of entire neurons (Fig. 2d) directly from local point cloud contexts. A low-dimensional UMAP (uniform manifold approximation and projection)[18] projection of the latent feature space led to clusters that contained known morphological neuron types of the analyzed tissue (zebra finch Area X), such as putative cholinergic and pallidal-like neurons. This analysis revealed additionally that Area X might harbor more cell types, for example local neurons that form synapses with excitatory ultrastructural characteristics (Fig. 2d, STN)—a neuron type in Area X that has so far only been physiologically identified[19] but not anatomically characterized. This shows that the dense morphology information collected from an electron microscopy (EM) connectomic dataset may eventually be as powerful for the characterization of neuron types in a brain area as single-cell gene expression data, while additionally containing full connectivity information.

The upgraded pipeline allowed us to extend an earlier analysis that relates the firing rates of striato-pallidal neuron classes to the mitochondrial content of different cellular compartments[7]. With access to thousands of synapses and their associated cell types, we tested whether larger synapses preferentially recruit mitochondria presynaptically, which could accommodate increased local energy demand[20]. As predicted, we observed that larger synapses were closer to mitochondria than smaller synapses, with a cell type-dependent distance distribution (Fig. 2e; distance to median lower half of synapses for medium spiny neurons (MSN) 0.833 μm; globus pallidal-like neurons (GP) 0.267 μm; median upper half MSN 0.339 μm, GP 0.232 μm; N synapses GP 7,482, MSN 59,131; $P = 0.0$ for lower versus upper half size population in both cell types using a two-sided Kolmogorov–Smirnov test). Pallidal-like neuron types, which exhibit high firing rates, showed a smaller synapse–mitochondria distance compared to sparsely firing striatal spiny neurons. Furthermore, synapse–mitochondria distance is similar for large and small GP synapses, whereas mitochondria appear to be recruited selectively to large MSN synapses (Fig. 2e and Extended Data Fig. 2). This analysis demonstrates that queryable EM connectomic datasets with dense ultrastructural annotation enable insights well beyond connectivity analyses, and future analyses based on the spatial distribution of ultrastructure might shed light on topics ranging from synaptic plasticity rules to neuromodulation.

Connectome data accessibility is a key issue, especially for researchers who have not originally produced and analyzed a VEM dataset. We therefore developed a web client for datasets processed with SyConn2, by building upon the Neuroglancer[21] interface (Fig. 2f). The web-based SyConn2 client allows neuroscientists to inspect a connectome without downloading all reconstructed neurons and synapses and to perform simple analyses (for example, visualizing neurons synaptically connected in a row) directly in the web browser, a feature that should help a larger research community find utility in rapidly emerging connectomic datasets.

## Online content

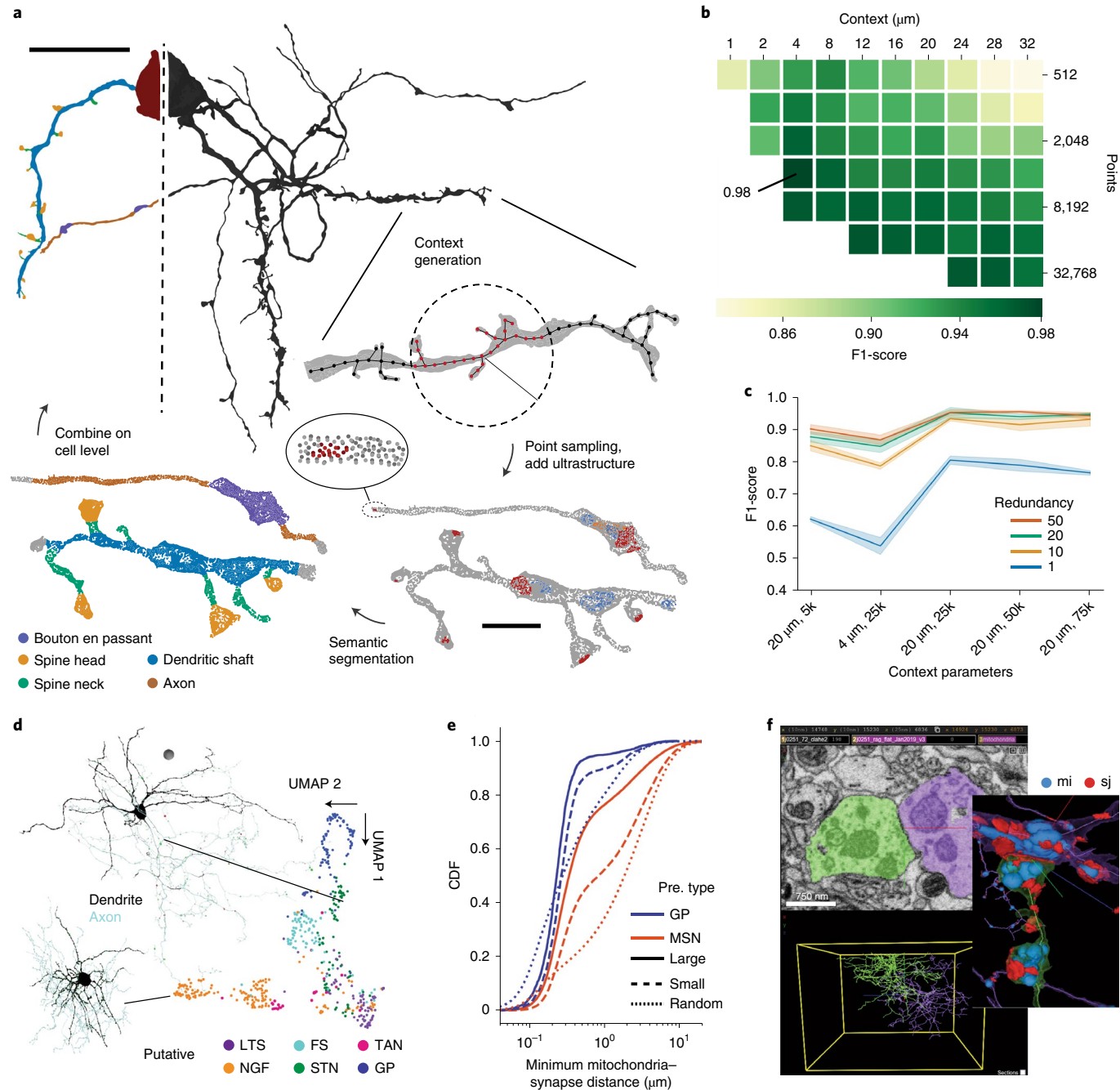

**Fig. 2 | SyConn2 processing and analyses of neuron reconstructions.**
**a**, Semantic segmentation of cell surfaces with point cloud neural networks. Surface points of the cell and ultrastructure within an input context were subsampled and presented to the model. Context predictions are then combined on the cell level. **b**, Grid search for optimal context parameters (radius, number of points) evaluated at synapse locations (88 spine head and 94 dendritic shaft) with weighted average F1-score (dendritic shaft, spine head and a combined axon and soma class). **c**, Classification performance of putative cell types dependent on the context and the number of bootstrapping samples (redundancy). For example, 20 μm, 5k refers to a 20 μm radius with 5,000 points. The confidence interval is mean ± standard deviation of three training repetitions for each parameter pair. **d**, UMAP dimensionality reduction of learned unsupervised latent space of 531 neurons in the dataset that contained soma, axon and dendrite

(MSNs not considered). LTS, low-threshold spiker; FS, fast-spiking interneuron; TAN, tonically active cholinergic neuron; NGF, neurogliaform interneuron; STN, subthalamic nucleus-like neuron; GP, pallidal-like neuron. Colors indicate putative cell type based on supervised classification. **e**, Cumulative distribution function (CDF) of the minimal distance between axo-dendritic synapses (and a random control) and mitochondria in GP and MSN split into small and large synapses (less than or equal to and greater than median of mesh area; median GP 1.16 μm², MSN 0.75 μm²; N synapses GP 7,482, MSN 59,131; see also Extended Data Fig. 2b for synapse size distributions; N random control locations: GP 37,149, MSN 6,128,974). Pre. type, presynaptic cell type **f**, Example of a GP–GP synapse visualized with the web-based SyConn2 client. Scale bars, 1 μm in EM section and 4 μm in renderings (**a**), 20 μm for the cell and 2 μm for the context (**b**) and 10 μm sphere radius (**d**).

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

## Methods

### Input segmentation maps and ultrastructure predictions

A cell instance segmentation map was generated by M.J. using flood-filling neural networks as reported earlier[4], with additional training data provided by annotators at the Max Planck Institute of Neurobiology and ariadne.ai AG. Synaptic junction (sj), sj type (symmetric and asymmetric), vesicle cloud (vc) and mitochondria (mi) voxel segmentation maps were also provided by M.J. using a 3D convolutional neural network model that predicts these classes on a per voxel level, followed by thresholding.

A myelin segmentation map (four-fold downsampled) was generated using SyConn's neural network model chunk inference pipeline, which divides the dataset into a configurable number of data chunks (used here: cube size of [482, 481, 236] voxels with additional [20, 30, 31] overlap on every side) to enable parallel processing. For the myelin inference, a model based on the 3D U-Net architecture[22] was used, which had the following parameters: 32 output channels in the first layer; output channels increase by a factor of two in every downpath layer; four downpath layers; first and third layer had a z-kernel extent of 1; ReLU activation; batch normalization. This model was implemented in elektronn3 (https://github.com/ELEKTRONN/elektronn3/), a training and inference framework that builds on PyTorch[23] and provides features for working with large-scale 3D image data.

### SegmentationObject generation

The (binary) input segmentation maps for mitochondria and vesicle clouds were transformed into an instance segmentation by a 3D watershed procedure (segmentation.watershed from the scikit-image package[24]), which was performed on the distance transform (filters.distanceTransform from the vigranumpy package, https://ukoethe.github.io/vigra/doc-release/vigranumpy/) of the input maps. The seeds for the watershed were generated from the morphologically modified (vc: binary opening, binary closing, binary erosion; mi: binary opening, binary closing, ×3 binary erosion) input maps using connected component analysis (ndimage.label from the scipy package[25]). Compute tasks were distributed across the workers by chunking (512 voxels edge length; 6, 2 voxels overlap for mi, vc). Chunk-wise identities (IDs) were made unique dataset-wide, and the overlap regions were used to unify IDs of objects that spanned multiple chunks. The resulting 3D connected components of voxels (supervoxels) were subsequently analyzed and stored in an accessible format, as described in the next paragraph.

The supervoxels formed the basis for SegmentationObjects (SO), which store additional properties (representative coordinate, voxel bounding box, voxel count, mesh, skeletons, mesh area and mesh bounding box) of cells, ultrastructure (mi, vc), contact sites (cs; Synapse–cell association), and synapse fragments and agglomerates (syn; Synapse–cell association) and are collected in SegmentationDatasets (SD), with separate SDs for each type. A SD is a key–value store that provides an interface to individual SOs. The SO property extraction was performed on 3D chunks (512 voxels edge length) of every ultrastructure's instance segmentation. In a single pass, the mesh, voxel count, bounding box and representative coordinate of all segmentation IDs in a cube were computed, and the partial results were merged in a final reduction step. For every syn object, the fraction of overlapping symmetric and asymmetric voxels was determined. Cell SOs also store the ID and fraction of overlapping ultrastructure segmentation voxels and were skeletonized using kimimaro[26]. Meshes of cells, mitochondria and vesicle clouds were computed with zmesh (https://github.com/seung-lab/zmesh).

### Synapse–cell association

We performed synapse identification through a multistep extraction process.

In a first step, a contact site instance segmentation was generated by iterating over the cell segmentation and storing adjacent supervoxel IDs. At every boundary voxel (6-connectivity) of the cell segmentation, a partner cell ID was identified by finding the majority ID within a window of [7,13] voxels (voxel size 10, 10, 25 nm). If a majority ID was found (background and the source boundary voxel ID were excluded), the contact site voxel was assigned a value that allowed the retrieval of the two partner cells (bit shift combination to uint64 in case of uint32 cell segmentation; tuple of uint64 in case of uint64 cell segmentation). The resulting thin boundary instance segmentation was morphologically closed ($N = 7$ iterations; this is sufficient to close the maximum distance of adjacent cells found through the adjacency filter) and dilated twice afterwards. Note that one instance in this segmentation represents all contact sites between a cell supervoxel pair, as the contact instance ID is the same, even if the supervoxels touch at different locations.

In a second step, synapse fragments and agglomerates on the supervoxel level (sv-syns) were generated by intersecting voxels of the sj foreground prediction and of the contact site instances. Individual putative synapses between two cells were obtained by computing connected components on a graph that was built with the voxels of sv-syns of all the cells' supervoxels that form such sv-syns between the cell pair. Within sv-syns between the same supervoxel pair, edges were added between voxels not farther apart than two voxels, and sv-syns of different supervoxel pairs were connected if their closest voxels were within a distance of at most 250 nm. For generating synapse meshes, the function 'create_from_point_cloud_poisson' from open3D[27] was applied on the voxels of the individual synapse objects. The resulting synaptic objects were further assigned a probability value using a random forest classifier ($N = 10$ features: synapse size in voxels, mesh area, numbers and voxel counts of presynaptic and postsynaptic mitochondria and vesicle clouds; trained on 300 putative synapse objects, manually annotated into 156 synaptic and 144 nonsynaptic), with 0 meaning least synaptic and 1 meaning most synaptic. The voxel count features for nearby (maximum representative coordinate distance of 4 μm) mi and vc objects were calculated by finding the number of mi or vc mesh vertices with a maximum distance of 2 or 1 μm to the synapse voxels, followed by dividing this vertex count by the total object vertex count to obtain a fraction that could then be multiplied by the object voxel count, resulting in the number used as features (mesh vertices and synapse voxels were 2-fold subsampled).

### SuperSegmentationObject generation

The SuperSegmentationObject (SSO) class was implemented to represent agglomerated cell reconstructions. Based on a supervoxel graph that defines which cell fragments belong to the same biological cell, an SSO aggregates the properties of the corresponding cell SOs (representative coordinate, bounding box, mesh, skeleton) and contains associated ultrastructure SO IDs and further analysis results (cell type predictions and certainties, vertex and skeleton node compartment prediction, local morphology embeddings, spine head volumes, myelination status).

SO properties were merged as follows. Representative coordinate: first SO representative coordinate; bounding box: minimum and maximum values of all SO bounding boxes; meshes: concatenation of vertices and indices; skeleton: concatenation of nodes and edges, adding edges between the closest skeleton nodes of skeleton fragments (resulting either from chunked processing or not agglomerated SOs) until the whole-cell skeleton was a single connected component.

Myelin predictions were mapped onto cell skeletons by storing the fraction of myelin voxels within a cube of size [11, 11, 5] voxels (voxel size (nm): 40, 40, 100) at every skeleton node and thresholding (per voxel probability threshold 0.5 and classification via majority vote). The node predictions were smoothed using a running majority vote on all neighboring nodes collected within a 10 μm path traversal starting from the source node.

### Context generation for point cloud processing

The reconstructed cells were split into regions of overlapping surface meshes (mesh contexts), controlled by parameters for vertex count

and context radius. This was done by choosing skeleton nodes as pivot locations around which a subgraph of adjacent nodes within a maximum distance (here called context radius) was constructed. For a fast lookup from skeleton node to mesh vertices, each skeleton node was assigned a set of mesh vertices by finding the nearest node for every vertex (Voronoi partitioning). The local mesh context corresponding to the pivot location was built by combining the vertices of all skeleton nodes in its subgraph.

Pivot nodes were spaced regularly on the cell skeleton until the full neurite was chunked into mesh contexts, either one or multiple times, depending on the chosen redundancy of mesh context generation. Meshes of ultrastructure (mi, vc) and synapses (plain or separated into excitatory and inhibitory) were combined with the neurite mesh and distinguished using a one-hot-encoded feature vector. Some experiments also included an additional binary input channel encoding the presence of myelin layers around axons. All meshes were downsampled by defining a voxel grid and selecting only one vertex per voxel (voxel edge length: cell 80 nm, mitochondria 100 nm, vesicle clouds 100 nm, synapses 100 nm; downsampling was performed by the 'voxel_down_sample' method of the open3D[27] PointCloud class) to standardize point densities and remove artifacts from the reconstruction process.

### Point cloud model training

Unless stated otherwise, models were trained until training loss convergence, using random point samples of the extracted mesh contexts, mini-batches, Adam optimizer (betas: 0.9, 0.999), cross-entropy loss, a stepwise learning rate decay and ReLU activation after each layer. All models were implemented using PyTorch[23] and LightConvPoint (https://github.com/valeoai/LightConvPoint) and trained via the elektronn3 framework.

Mesh contexts used as training samples were transformed by multiple point cloud augmentations. These augmentations consisted of random noise added to the point positions, random rotations and flipping, elastic transformations[28] and anisotropic scaling. All point cloud processing methods were implemented in the MorphX package.

### Semantic segmentation of dendrites

For the surface segmentation of dendrites into dendritic shaft, spine neck and spine head, we applied a hierarchy of two models. The coarse-level model was used to separate dendrite from axon and soma, and the predictions of the second model further distinguished the dendritic parts into dendritic shaft, spine neck and spine head. Both high-level (classes: 2, dendrite versus a combined axon and soma class) and fine-level models (classes: 3, dendritic shaft versus spine neck versus spine head) were trained and tested on the ground truth of the high-resolution surface segmentation task from ref. [14].

To analyze the effects of point number and context radius on the dendritic inference task (one-dimensional input features using only cell surface points), we conducted a grid search varying these two parameters while keeping the results of the coarse-level morphology model (input parameters: 15,000 points, 15 μm context, four-dimensional input features using one-hot encoding of cell, mi, vc and synapse points) fixed. For the coarse-level model, the architecture was the same as the one used for the fine-level model with more than 2,048 input points (see below), and predictions were performed on cell surface points only, excluding vertices of ultrastructure (mi, vc) and synapses.

For the grid search of the dendritic model, we only generated matrix entries in which most mesh contexts would still hold more points than requested by the point sampling. In the case that the number of points in the extracted mesh context was fewer than the requested volume, the missing points were randomly sampled from the original set of points. Each cell in the training set was split five times. We used four different architectures, depending on the point number. All architectures used kernels with 16 points each. For matrix entries with 512 points, we used architectures with the following layer specifications: (1: 32 kernels, 32 neighbors, no reduction), (2: 32, 32, reduction to 256 points), (3: 64, 32, reduction to 64 points), (4: 64, 16, 16), (5: 64, 8, 8), (6: 64, 4 deconvolution, deconv, to 16, residual to 5), (7: 64, 4 deconv to 64, residual to 4), (8: 32, 8, deconv to 256, residual to 3), (9: 32, 16, deconv to original point cloud, residual to 2), (10: fully connected shared across all points, residual to 1). Two more layers between layer 1 and 2 and layer 8 and 9, respectively, were added for 1,024 input points: (1 and 2: 32, 32, reduction to 512), (8 and 9: 32, 16, deconv to 512 + residual). For 2,048 points, two layers (additional to the 1 and 2, 8 and 9 layers) were added: (1 and 2: 32, 32, reduction to 1,024), (10 and 11: 32, 16, deconv to 1,024 + residual). Models with more than 2,048 input points shared the same architecture as for 2,048 points but changed the reduction pathway to no reduction, 2,048, 1,024, 256, 64, 16, 8. The total number of trainable parameters was in the range from 541,603 to 593,699, depending on the model architecture, as described above.

All models used GroupNorm[29] after each layer (except the fully connected ones). The point cloud reduction was done by efficient point sampling with space quantization[30]. All fine-level morphology models were trained until convergence (after 1,400–3,000 epochs, training time from 4 h to 30 h, training speed from 3.1–1.4 samples per second) with batch sizes 32 (fewer than 2,048 points), 16 (fewer than 8,192), 8 (fewer than 16,384) and 4 (fewer than 32,768) using an initial learning rate of $1 \times 10^{-3}$ (scheduler step size of 1,000, decay 0.99). The coarse-level morphology model was trained using a batch size of 4, DiceLoss (class weights dendrite: 2, combined axon and soma class: 1), AdamW optimizer and an initial learning rate of $2 \times 10^{-3}$ (scheduler step size of 100, decay of 0.996); input points were normalized to a unit sphere.

Model performances were evaluated on a set of manually labeled synapses in four neuron reconstructions (94 on dendritic shaft and 88 on spine head; the same as in ref. [14]). These neurons were split five times with different context locations and processed by the coarse-level and all fine-level models. Vertices with multiple predictions (for example, because they were part of multiple mesh contexts) were assigned the result of a majority vote on all their predictions. The final synapse label was found by majority vote on the predictions of the 20 closest vertices with respect to the representative coordinate of the synapse. Each matrix entry presents the mean weighted (by synapse support) average F1-score of three fine-level models with the same architecture and input settings, but trained with different random seeds.

### Cell type classification

For the supervised type classification of neurites, 253 neuron reconstructions were manually labeled by an expert, not necessarily covering all distinguishable cell types of this brain area (number of labeled classes: 11). These included three interneuron classes (putative low-threshold spiking interneuron (LTS), putative fast-spiking neuron (FS) and putative neurogliaform interneuron (NGF) in Fig. 2d) forming inhibitory synapses and one local neuron class with excitatory synapses (putative excitatory subthalamic nucleus-like (STN) in Fig. 2d). The ground truth was split into training and test data using 10-fold cross validation. Each split was used to train three models, each starting with a different random seed for training batch generation and initial weights to estimate the model variance. The context generation was parameterized by radius and number of points (Context generation for point cloud processing), and seed nodes were sampled uniformly. Vertex features were represented via a six-dimensional one-hot encoding of mitochondria, vesicle clouds, inhibitory and excitatory synapse, and myelinated and unmyelinated cell surface. The myelination information was propagated from the skeleton node associated with a cell surface vertex (Voronoi partitioning). Input point coordinates were centered and scaled by 10% of the context radius.

Model architecture: five ConvPoint layers each using 16 kernel elements; group normalization before swish activation[31] with the following parameters (number of output channels, reduction to N points,

*k* nearest neighbors): (64, 4,096, 32), (128, 1024, 32), (256, 512, 16), (256, 256, 16), (512, 128, 16). The resulting 512 features were averaged across the anchor 128 points. An additional dropout (rate 0.3) was applied before the final two fully connected layers with 128 and 11 output channels. Convolutions with point reduction used heuristic point sampling[15].

The default training configuration was modified as follows: initial learning rate $5 \times 10^{-4}$, learning rate scheduler step size 100, and decay of 0.99. To speed up the data preparation during training, a single batch (batch size 10) contained random contexts of only one cell reconstruction. Parameter updates were performed after accumulating gradients of 10 batches to improve the learning signal.

For the whole dataset inference, we used $N = 20$ editions each with 50,000 points and a context radius of 20 μm for the type classification of a neuron reconstruction. During inference a fixed number of seed nodes was used, and the resulting per-class logits were accumulated and normalized to 1. The resulting pseudo-probabilities $p_i$ for each class, indexed by $i$ (with $C$ denominating all classes) were used for classification of the cell type (class with the maximum probability) and to calculate a certainty estimate of the prediction based on its entropy $H$:

$$\text{certainty} = 1 - H/H_{\max} = 1 + \frac{1}{H_{\max}} \sum_{i=1}^{C} p_i \log_2 p_i = 1 + \sum_{i=1}^{C} p_i \log_C p_i \text{ with}$$

$$H_{\max} = -\sum_{i=1}^{C} \frac{1}{C} \log_2 \frac{1}{C} = \log_2 C.$$

### Self-supervised cell embeddings

Furthermore, we trained a model (same architecture as for the supervised task; context: 15 μm and 25,000 points; ten-dimensional (10D) output) via triplet loss[32] to embed the morphology of two proximal locations of the same cell (first context location drawn randomly from all cell skeleton nodes; second context center drawn uniformly within 15 μm distance along the cell skeleton) closer in a 10D latent space than a cutout of a different cell (drawn randomly). This self-supervised training procedure did not require any additional manual annotations and was performed on all sufficiently large neuron reconstructions (SSO). Neurons (or fragments) that had a bounding box diagonal less than two times the input context of the model (less than 30 μm) were excluded. The local embeddings (represented by their source nodes used for context generation, termed context center) were aggregated to cell level by calculating their mean within the same compartments (axon, dendrite) and adding the two resulting vectors. Context centers of a cell were generated using voxel downsampling of the mesh vertices with a voxel size of half the context size and drawn randomly.

Training configuration: Initial learning rate $5 \times 10^{-4}$; learning rate scheduler step size 250 and decay of 0.995; margin ranking loss with a margin of 0.2 and a batch size of 16. Every cell skeleton node was assigned the morphology embedding vector associated with the spatially closest context center.

We only considered cell reconstructions with a soma skeleton length more than 10 μm, axon and dendrite skeleton lengths more than 200 μm, and those that were additionally not classified as MSN or an axon class only projecting to Area X for the unsupervised cell type analysis (Fig. 2d), to focus the embedding on the rare cell types of Area X. Overall, 531 cells passed these criteria, and for each cell we constructed a compound 10D latent space by averaging the local triplet-loss embeddings generated at cell skeleton nodes along each embedding dimension for the axon and dendritic compartments separately, followed by summation of the two vectors. These 531 10D vectors were then reduced to two dimensions with the following UMAP[18] parameters: n_neighbors=60, metric='euclidean', random_state=0, min_dist=0.05, n_epochs=1,000.

### Analysis of the minimal mito–synapse distances

The minimal distances between presynaptic MSN and P (predicted GPi and GPe combined) synapses and mitochondria were calculated as the Euclidean distance between a representative synapse coordinate and the closest mesh vertex (point on the surface; downsampled to a voxel size of 200 nm) of the cell's mitochondria. Cells were filtered as follows: minimum axon, dendrite and soma path length of 100 μm, 50 μm and 5 μm, respectively, and cell type certainty (definition above) of at least 0.75. Only axo-dendritic synapses with a probability (random forest classifier; Synapse–cell association) above 0.8 were included. Path lengths were calculated by summing the edges between cell skeleton nodes that were labeled as the respective compartment type. For this analysis, the compartment predictions were performed with the same model that was used in ref. [14] for spine predictions (spine head, spine neck, dendritic shaft, combined axon and soma class) and a second model using the same architecture for larger structures and axonal compartments (dendrite, soma, axon, bouton *en-passant*, terminal bouton; context size: 40.96 μm × 20.48 μm × 40.96 μm captured with three renderings per location at a resolution of 1,024 by 512 pixels; rendering locations were sampled using a voxel downsampling of the mesh vertices with a voxel size of 13.65 μm; trained on 45 manually labeled reconstructions). Vertex predictions were propagated to skeleton nodes by calculating the majority vote of the *k* nearest prediction locations (compartments with $k = 50$, separately stored for the two models) and which were in turn smoothed using a running majority vote on all neighboring nodes collected within a 10 μm path traversal starting from the source node.

The control for the minimal syn–mito distances was performed by sampling locations on the cell's axonal compartment surface randomly and calculating the distance to the closest mitochondria mesh vertex (downsampled to a voxel size of 200 nm). For each cell, up to 1,000 skeleton nodes that belonged to the axon (fewer if the cell contained fewer nodes) were drawn. For each node a random vertex from all cell mesh vertices, that were assigned to that node via Voronoi partitioning, was chosen as the control location.

The two-sided Kolmogorov–Smirnov test (using the ks_2samp method from the scipy package in 'asymp' mode and with alternative='two-sided') returned *P* values of 0.0 for lower versus upper half size population (split using median) for GP (test statistic 0.154) and MSN (test statistic 0.245) and for lower versus control for GP (test statistic 0.195) and MSN (test statistic 0.206); *N* synapses GP 7,482 and MSN 59,131; *N* random control locations: GP 37,149 and MSN 6,128,974.

A manual synapse assessment was performed by J.K. and P.S. on 52 randomly selected synapse objects (12 GP and 13 MSN from the lower half of synapse area distributions, and 13 GP and 14 MSN of the upper half; 51 were classified as true synapse, 1 upper MSN as false). The synapse objects were selected from a subset of 100 randomly selected MSN and 38 GP cells. The assessment was performed blind; that is, it was hidden during the annotation as to which cell type and synapse size a synapse belonged to.

### Cost estimation

The timing experiments were performed with a dynamically created SLURM cluster on the Google Cloud Platform using elasticluster (https://github.com/elasticluster/elasticluster). In total, 24 compute nodes (n1-highmem-32), each with 2 Tesla P100 GPUs, 32 virtual cores and 208 GB RAM, were used in combination with a Gluster filesystem (https://www.gluster.org/, four server nodes with SSD) and a 10 TB persistent disk to store the input data (aligned EM data, cell segmentation, myelin, sj, mi, vc and synapse type predictions). The timed processing steps were grouped into CPU-only (data store, synapse extraction, synapse enrichment; 68.56 h at 1.812 teravoxels) and CPU + GPU (morphological analysis; 8.44 h point based, 27.51 h multiview based at 1.812 teravoxels). See Supplementary Text 3 and Extended Data Fig. 1 for details about timings and executed steps. Assuming GPU nodes only for GPU-relevant processing steps, the cost per teravoxel for the different categories summed to approximately US$1,200 for CPU-only (US$1.325

hourly rate for one CPU node, based on https://cloud.google.com/products/calculator), US$380 for GPU + CPU (point-based; US$1,200 for multi-view models; US$3.36 hourly rate for one GPU node) and US$260 for infrastructure (US$4.84 hourly rate for persistent disk and US$0.348 per file system server node); in total US$1,840 per teravoxel.

## Reporting summary

Further information on research design is available in the Nature Research Reporting Summary linked to this article.

## Data availability

All datasets are available at syconn.esc.mpcdf.mpg.de and licensed under the Creative Commons Attribution 4.0 International CC-BY license. Source data are provided with this paper.

## Code availability

All source code, implemented in Python 3.7, is available on GitHub: SyConn2: https://github.com/StructuralNeurobiologyLab/SyConn (GPL-2.0 license); MorphX: https://github.com/StructuralNeurobiologyLab/MorphX/ (GPL-2.0 license); elektronn3: https://github.com/ELEKTRONN/elektronn3/ (MIT license).

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

## Acknowledgements

We thank W. Denk for enabling this work in his department and feedback on the manuscript, H. Baier for manuscript feedback, E. Perlman for help with interfacing SyConn2 and Neuroglancer, J. Maitin-Shepard for developing Neuroglancer, and Google Research for providing cloud computing resources. We would also like to thank C. Guggenberger and his team at the MPCDF computing facility in Garching, Germany for support.

## Author contributions

P.S., S.D., J. Klimesch and J. Kornfeld designed and implemented SyConn2, with code contributions from F.S.. H.A. and A.M. implemented the SyConn2 web-client based on Neuroglancer. M.J. and V.J. contributed the FFN segmentation and other input segmentation maps. P.S. and J. Kornfeld wrote the manuscript, with contributions from all other authors.

## Funding

Funding was provided by NIH grant no. RF1 MH117809-01 (M.S.F., J.K.) and the Max Planck Society. Open access funding provided by Max Planck Society.

## Competing interests

F.S. and J.K. disclose financial interests in ariadne.ai ag. S.D., M.J. and V.J. are employees of Google LLC, which sells cloud computing services. The remaining authors declare no competing interests.

## Additional information

**Extended data** is available for this paper at https://doi.org/10.1038/s41592-022-01624-x.

**Correspondence and requests for materials** should be addressed to Joergen Kornfeld.

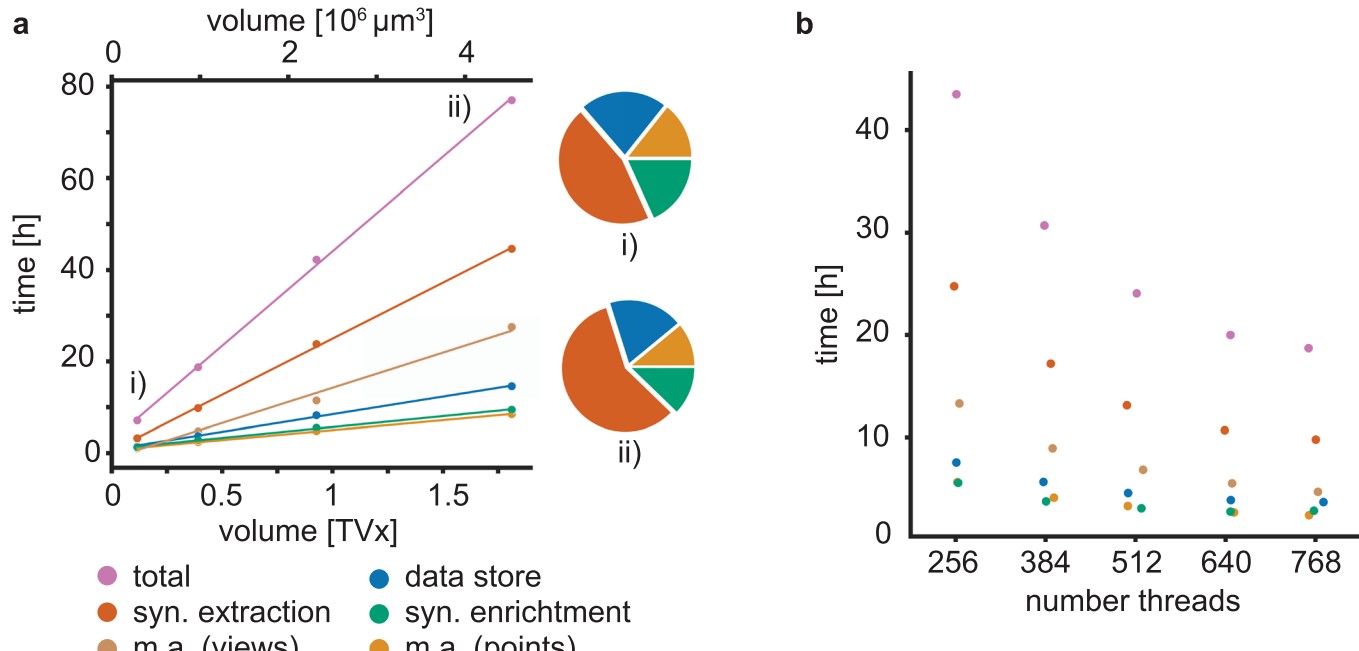

**Extended Data Fig. 1 | Timings of the different pipeline steps.** Timings are grouped into synapse extraction, data store, synapse enrichment and morphology analysis (m.a.) with multi-views (views) and point clouds (points). **a** Compute time as a function of the processed volume (in teravoxels, TVx). Pie charts show the fraction of the different steps relative to the total time at the smallest and largest test cube (i: 0.29 million µm³, syn. extraction: 0.45, data store: 0.22, syn. enrichment: 0.18, m.a. (points): 0.14; ii: 4.53 million µm³, 0.58, 0.19, 0.12, 0.11). The 'views' step was excluded for the 'total' timings and the pie charts (i, ii). Compute resources: 24 google cloud computing nodes (n1-highmem-32), each with 32 virtual cores (threads), 2 Tesla P100, 208 GB memory. **b** Compute time as a function of the number of available compute nodes (8, 12, 16, 20, 24). Processed volume: 0.391 teravoxels. Color code as in **a**.

**a**

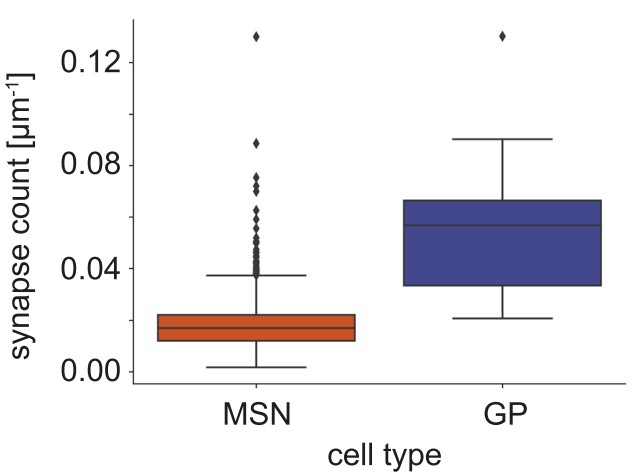

**b**

**Extended Data Fig. 2 | GP and MSN synapse properties. a** Box plot (median, lower and upper quartile; whiskers, 1.5x interquartile range above upper and below lower quartile; points, outlier) of the average synapse count per micrometer for cell types MSN (N = 6327, median: 0.017 μm⁻¹, Q1: 0.012 μm⁻¹, Q3: 0.022 μm⁻¹) and GP (N = 38, 0.057 μm⁻¹, 0.033 μm⁻¹, 0.066 μm⁻¹). Two-sided Mann-Whitney U test statistic: -9.71 and p-value: 2.57e-22. **b** Histogram of synapse mesh area (N synapses GP: 7,482, MSN: 59,131).

# Reporting Summary

## Statistics

For all statistical analyses, confirm that the following items are present in the figure legend, table legend, main text, or Methods section.

| n/a | Confirmed | |
|---|---|---|
| ☐ | ☒ | The exact sample size (*n*) for each experimental group/condition, given as a discrete number and unit of measurement |
| ☒ | ☐ | A statement on whether measurements were taken from distinct samples or whether the same sample was measured repeatedly |
| ☐ | ☒ | The statistical test(s) used AND whether they are one- or two-sided *Only common tests should be described solely by name; describe more complex techniques in the Methods section.* |
| ☒ | ☐ | A description of all covariates tested |
| ☒ | ☐ | A description of any assumptions or corrections, such as tests of normality and adjustment for multiple comparisons |
| ☐ | ☒ | A full description of the statistical parameters including central tendency (e.g. means) or other basic estimates (e.g. regression coefficient) AND variation (e.g. standard deviation) or associated estimates of uncertainty (e.g. confidence intervals) |
| ☐ | ☒ | For null hypothesis testing, the test statistic (e.g. *F*, *t*, *r*) with confidence intervals, effect sizes, degrees of freedom and *P* value noted *Give P values as exact values whenever suitable.* |
| ☒ | ☐ | For Bayesian analysis, information on the choice of priors and Markov chain Monte Carlo settings |
| ☒ | ☐ | For hierarchical and complex designs, identification of the appropriate level for tests and full reporting of outcomes |
| ☒ | ☐ | Estimates of effect sizes (e.g. Cohen's *d*, Pearson's *r*), indicating how they were calculated |

*Our web collection on statistics for biologists contains articles on many of the points above.*

## Software and code

Policy information about availability of computer code

| Data collection | No software has been used for data collection. |
|---|---|
| Data analysis | SyConn 2.0 (https://github.com/StructuralNeurobiologyLab/SyConn),  MorphX 0.1 (https://github.com/StructuralNeurobiologyLab/MorphX), python 3.7, slurm 20.02.6, cuda 10.2, pytorch 1.8, open3d 0.9.0, LightConvPoint 0.2, kimimaro 3.0.0, zmesh 0.5.1, elektronn3 alpha, scikit-image 0.18.3, vigra 1.11.1, scipy 1.6.3, elasticluster 1.3.dev28, Neuroglancer 2.19 |

For manuscripts utilizing custom algorithms or software that are central to the research but not yet described in published literature, software must be made available to editors and reviewers. We strongly encourage code deposition in a community repository (e.g. GitHub). See the Nature Portfolio guidelines for submitting code & software for further information.

## Data

Policy information about availability of data

All manuscripts must include a data availability statement. This statement should provide the following information, where applicable:

- Accession codes, unique identifiers, or web links for publicly available datasets
- A description of any restrictions on data availability
- For clinical datasets or third party data, please ensure that the statement adheres to our policy

Due to large storage requirements, the data sets cannot be made available in a repository but are available at syconn.esc.mpcdf.mpg.de for online viewing and inspection. Please contact the authors to obtain a copy upon reasonable request.

# Field-specific reporting

Please select the one below that is the best fit for your research. If you are not sure, read the appropriate sections before making your selection.

☒ Life sciences    ☐ Behavioural & social sciences    ☐ Ecological, evolutionary & environmental sciences

For a reference copy of the document with all sections, see nature.com/documents/nr-reporting-summary-flat.pdf

# Life sciences study design

All studies must disclose on these points even when the disclosure is negative.

| | |
|---|---|
| Sample size | The sample sizes for the analysis of the minimal mito-synapse distances included all neuron reconstructions from the data set as described in the methods. |
| Data exclusions | Neuron reconstructions were excluded based on minimum compartment size filters as described in the methods. |
| Replication | The mito-synapse distance analysis was performed on a single connectomic data set and were selected from 100 randomly selected MSN and 38 GP cells. |
| Randomization | The order of the synapses for the manual evaluation in the mito-synapse distance analysis was shuffled. |
| Blinding | The annotation of synapses during synapse evaluation for the analysis of the minimal mito-synapse distances was done without knowledge about associated groups (lower/upper half, cell type). |

# Reporting for specific materials, systems and methods

We require information from authors about some types of materials, experimental systems and methods used in many studies. Here, indicate whether each material, system or method listed is relevant to your study. If you are not sure if a list item applies to your research, read the appropriate section before selecting a response.

## Materials & experimental systems

| n/a | Involved in the study |
|---|---|
| ☒ | ☐ Antibodies |
| ☒ | ☐ Eukaryotic cell lines |
| ☒ | ☐ Palaeontology and archaeology |
| ☒ | ☐ Animals and other organisms |
| ☒ | ☐ Human research participants |
| ☒ | ☐ Clinical data |
| ☒ | ☐ Dual use research of concern |

## Methods

| n/a | Involved in the study |
|---|---|
| ☒ | ☐ ChIP-seq |
| ☒ | ☐ Flow cytometry |
| ☒ | ☐ MRI-based neuroimaging |

