## [Peer Review File · Nature Methods]

Peer Review Information

Manuscript Title: SyConn2: Dense synaptic connectivity inference for volume EM

Corresponding author name(s): Joergen Kornfeld

Reviewer Comments & Decisions:

Decision Letter, initial version:

Dear Joergen,

Thank you for your patience. Your Brief Communication, "SyConn2: Dense synaptic connectivity inference for volume EM", has now been seen by two reviewers. As you will see from their comments below, although the reviewers find your work of considerable potential interest, they have raised a number of concerns. We are interested in the possibility of publishing your paper in Nature Methods, but would like to consider your response to these concerns before we reach a final decision on publication. We therefore invite you to revise your manuscript to address these concerns.

- * include a point-by-point response to the reviewers and to any editorial suggestions
- * please underline/highlight any additions to the text or areas with other significant changes to facilitate review of the revised manuscript
- * address the points listed described below to conform to our open science requirements

* ensure it complies with our general format requirements as set out in our guide to authors at www.nature.com/naturemethods

* resubmit all the necessary files electronically by using the link below to access your home page

[Redacted] This URL links to your confidential home page and associated information about manuscripts you may have submitted, or that you are reviewing for us. If you wish to forward this email to co-authors, please delete the link to your homepage.

We hope to receive your revised paper within 4 weeks. If you cannot send it within this time, please let us know. In this event, we will still be happy to reconsider your paper at a later date so long as nothing similar has been accepted for publication at Nature Methods or published elsewhere.

OPEN SCIENCE REQUIREMENTS

REPORTING SUMMARY AND EDITORIAL POLICY CHECKLISTS

DATA AVAILABILITY

We strongly encourage you to deposit all new data associated with the paper in a persistent repository where they can be freely and enduringly accessed. We recommend submitting the data to discipline-specific and community-recognized repositories; a list of repositories is provided here:

<http://www.nature.com/sdata/policies/repositories>

All novel DNA and RNA sequencing data, protein sequences, genetic polymorphisms, linked genotype and phenotype data, gene expression data, macromolecular structures, and proteomics data must be deposited in a publicly accessible database, and accession codes and associated hyperlinks must be provided in the “Data Availability” section.

Please include a “Data availability” subsection in the Online Methods. This section should inform readers about the availability of the data used to support the conclusions of your study, including accession codes to public repositories, references to source data that may be published alongside the paper, unique identifiers such as URLs to data repository entries, or data set DOIs, and any other statement about data availability. At a minimum, you should include the following statement: “The data that support the findings of this study are available from the corresponding author upon request”, describing which data is available upon request and mentioning any restrictions on availability. If DOIs are provided, please include these in the Reference list (authors, title, publisher (repository name), identifier, year). For more guidance on how to write this section please see: <http://www.nature.com/authors/policies/data/data-availability-statements-data-citations.pdf>

CODE AVAILABILITY

Please include a “Code Availability” subsection in the Online Methods which details how your custom code is made available. Only in rare cases (where code is not central to the main conclusions of the paper) is the statement “available upon request” allowed (and reasons should be specified).

MATERIALS AVAILABILITY

ORCID

Nature Methods is committed to improving transparency in authorship. As part of our efforts in this direction, we are now requesting that all authors identified as ‘corresponding author’ on published papers create and link their Open Researcher and Contributor Identifier (ORCID) with their account on the Manuscript Tracking System (MTS), prior to acceptance. This applies to primary research papers only. ORCID helps the scientific community achieve unambiguous attribution of all scholarly contributions. You can create and link your ORCID from the home page of the MTS by clicking on ‘Modify my Springer Nature account’. For more information please visit www.springernature.com/orcid.

Best regards,
Nina

Nina Vogt, PhD
Senior Editor
Nature Methods

Reviewers' Comments:

Reviewer #1:

Remarks to the Author:

Summary: the manuscript describes SyConn2, a software toolkit for generating cellular and subcellular instances from segmentation of electron microscopy volumes of neural tissue. These outputs are integrated into a joint semantic representation, allowing a range of connectivity, cell type and compartment analyses. This tool is an update on the previously published SyConn, with the stated aim of being computationally more efficient and improving data access for researchers. On the former, this was partially done by replacing rendered views with point cloud representations of segmentation as input to morphology and type classification. On the latter, researchers are able to access reconstructed neurons, synapses and other segmented subcellular structures via a web browser (<https://syconn.esc.mpcdf.mpg.de/>).

Originality: the point cloud representations mentioned above and their use for cell typing analysis via unsupervised clustering is a novel implementation. Its reduced compute time, power and cost (shown in detail in figure S1 and lines 34-38, 399-414) could enable other experienced researchers to implement these methods. The evidence to show that this method can enable accurate cell typing and that could be applied more widely is however not very strong from the one example provided (figure 1e), using some quite broad cell classes (int1-3). The manuscript doesn't address if the point cloud representation is better than the previous views approach, therefore the novelty would be constrained to the performance of the method, not accuracy of output.

As for other novel aspects of this toolkit, it is difficult to gather what updates have been implemented, regarding the authors' prior publications about the toolkit and method (references 11 & 18) and to other tools cited for extracting and analyzing synaptic connectivity.

Data and Methods: although the methods are described extensively, as mentioned above, it is often unclear what particular improvements or differences were implemented with respect to referenced works. If the performance of the point cloud representation is the major aspect of novelty and significance of this toolkit, this may be better communicated by replacing either figure 1c or d with supplementary figure 1a. Regarding the significance of the methods, 2 examples are shown (figure 1e and f). Comments on the cell typing are given above. Figure 1f, together with supplementary figure 2 give an illustrative exploration of inter-type differences for the distribution of mitochondria to synapses. Adding the synaptic size distributions for both cell types would help the reader to judge whether the difference shown might simply be an effect of synaptic size rather than mitochondria to synapse distance.

The evidence to show that this method can enable accurate cell typing and could be applied more widely is however not very strong from the one example provided (figure 1e). It is not clear from which point cloud sampling-based embeddings are learned to allow semi-supervised cell typing. Regarding data availability, a Neuroglancer-based browser allows access to neuron reconstructions labelled by type and subcellular compartments, including synapse junctions. Exploring the 2D and 3D data in the Neuroglancer framework is intuitive and already familiar to many users, thus a good choice for sharing the data more widely.

Conclusions and suggested improvements: the review above already suggests a number of major improvements.

Minor points:

- Citation styles are mixed, and not in a way consistent with inline versus note citations. For example, line 284 cites author inline, but others in same paragraph do not.
- Some section references do not correspond to section titles. For example, multiple references are made to “synapse extraction”, but actual methods section title is “synapse-cell association”.
- acronyms are often not written in full when first shown in the text. For example MSN and GP (line 83). Many acronyms are also not written in full in the figure legend.
- Figure 1: inconsistent cell type coloring between panels e, f
- Figure 1b: en passant not “en passent”
- line 130: “equally provided” should likely be “also provided” or simply “provided”
- line 145: “(binary)input” missing a space
- line 168: As written, this citation does not need to be inline/long-form.
- line 195: What is this RF trained on? The same ground truth for junction detection?
- line 340: “Neurons(-fragments)” missing a space

Clarity and context: one point regarding clarity: in line 97, it is stated that simple analyses can be done directly on the SyConn web-browser. It is not clear what type of analyses are intended, as this is a visualisation and exploration tool. Adding a brief example would be helpful.

Reviewer #2:

Remarks to the Author:

Key results

The paper propose SyConn2, a cost-effective automated pipeline that can generate annotated connectome from input image volumes and do morphological analysis of the structures. Building upon previous work on automatic neuron reconstruction [1], the paper extend the work to reconstruct neurons with its ultrastructures (synaptic junctions, vessicles and mitchondria) and part segmentations (spine head, spine neck, axon, and etc.). Point cloud representation is used when doing part segmentation of neurons as well as cell classification (both under supervised and unsupervised setting). Finally, the reconstructed neurons are used to perform an analysis of the minimal mito-synapse distances.

From this, two key contributions can be identified.

1. Reconstruction of neurons with its ultrastructures and parts.
2. Using point cloud representation for part segmentation and cell classification.

Significance

This can greatly aid the research in connectome analysis. Even though there is previous work like [1,2] that perform segmentation of different structures, this work extract and combines different parts in a neuron as a single object (SuperSegmentationObject). This greatly improves the type of analysis that could be done.

Data and methodology

Image data used are standard EM data used for connectome analysis.

The proposed pipeline is based on stablished deep learning approaches for analysing image volumes and point clouds.

Main claims in the paper are supported by quantitate analysis. Results on cell type classification and cell part segmentation (subcellular compartment models) is presented in supplementary materials (Supp.

Text 1, 2). Cost analysis is presented in the paper (line 399). Some important experiments related to hyper-parameter tuning is also presented (paragraph starting from line 258).

To capture the uncertainty in the model, in Supp. Text 1. 10 fold cross validation is used and mean/std of the metrics are reported. In Supp. Text 2. average over 10 model checkpoints is reported.

Qualitative results can be found at <https://syconn.esc.mpcdf.mpg.de/>

Overall, we can observe very good neuron reconstructions. But it has to be noted that we can also see a noticeable amount of split errors (small false-positive regions inside/next-to neurons).

Conclusion

Qualitative results (<https://syconn.esc.mpcdf.mpg.de/>) present the best evidence towards the robustness, validity and reliability of the approach since its evaluated on the whole volume. Even though we observe some errors in the segmentations, overall pipeline is capable of successfully generating annotated connectome.

Suggested improvements / Unclear statements

Here I list several points (in order of their appearance in the text)

Line 29:

The term 'SyConn2' is introduced in line 29. Then in line 34, 39, 42 the term 'SyConn' is used and is referring to 'SyConn' proposed in [2]. But this is not explicitly introduced in the paper. Therefore, it would be better if 'SyConn' is also defined clearly.

Paragraph starting at line 133:

Statement 1: "A myelin segmentation map (4-fold downsampled) was generated using SyConn's neural network model chunk inference pipeline..."

Statement 2: "For the myelin inference, a model based on the UNet architecture was used, which had the following parameters..."

Is the U-Net part of SyConn? If SyConn can produce myelin segmentation maps (according to statement 1), why a U-Net is necessary?

Line 149:

For generating seeds, morphologically modified input segmentation maps has been used. Are these modified maps used as the final segmentation as well? If yes, why can we see small false negative regions (holes) in segmentation maps, for instance we can observe them inside the green and purple neurons in Fig 1.h. If not, why are they not used?

Line 221:

In this paragraph, use of point cloud representation is introduced. Why was point cloud representation selected over mesh representation? Mesh representation can better capture the shape and there are deep learning architectures that can process meshes similar to point clouds. Therefore, it might have produced competing or superior performance compared to the point cloud representation.

Paragraph starting at line 351:

How is the feature embedding for the unsupervised classification of cells computed? Based on this paragraph, the same network trained during supervised classification is used here. If that's the case, it's better to use the term 'pre-trained network' instead of 'unsupervised'. If it's not the case, the methodology used to train the network that produce the embedding is not clear.

Line 361:

In this paragraph, an example of an analysis that could be performed with the SyConn2 connectome results is presented. It would be interesting, if authors can discuss such other avenues the connectome data produced by SyConn2 could be used (no experiments required).

References in the paper

Paper cites recent publications.

Minor suggestion:

Line 137: Citing 3D U-Net [3] is more accurate.

Reviewer's expertise

My expertise is in machine learning/computer vision.

References

1. Januszewski, M. et al. High-precision automated reconstruction of neurons with flood-filling networks. Nat. Methods 15, 605–610 (2018).
2. Dorkenwald, S. et al. Automated synaptic connectivity inference for volume electron microscopy. Nat. Methods 14, 435–442 (2017).
3. Çiçek O, et al. 3D U-Net: Learning Dense Volumetric Segmentation from Sparse Annotation. MICCAI (2016).

Author Rebuttal to Initial comments

Reviewer #1

Remarks to the Author:

Summary: the manuscript describes SyConn2, a software toolkit for generating cellular and subcellular instances from segmentation of electron microscopy volumes of neural tissue. These outputs are integrated into a joint semantic representation, allowing a range of connectivity, cell type and compartment analyses. This tool is an update on the previously published SyConn, with the stated aim of being computationally more efficient and improving data access for researchers. On the former, this was partially done by replacing rendered views with point cloud representations of segmentation as input to morphology and type classification. On the latter, researchers are able to access reconstructed neurons, synapses and other segmented subcellular structures via a web browser (<https://syconn.esc.mpcdf.mpg.de/>).

Originality: the point cloud representations mentioned above and their use for cell typing analysis via unsupervised clustering is a novel implementation. Its reduced compute time, power and cost (shown in detail in figure S1 and lines 34-38, 399-414) could enable other experienced researchers to implement these methods.

We would like to thank the reviewer for the positive feedback.

The evidence to show that this method can enable accurate cell typing and that could be applied more widely is however not very strong from the one example provided (figure 1e), using some quite broad cell classes (int1-3).

After exploring the songbird data more, we are now confident to directly use more fine-grained labels for the broad int classes and exc class, and separated them into putative excitatory subthalamic nucleus-like (STN), putative low-threshold spiking interneuron (LTS), putative fast-spiking neuron (FS) and putative neurogliaform interneuron (NGF) types. We agree in general that SyConn2 and EM-based connectomics in general has to demonstrate its power for accurate cell typing further, especially in comparison to single cell RNA sequencing, but think that such experiments on many data sets are out of scope for this brief communication. This is also the reason why we used somewhat conservative language in the main text: 'This shows that the dense morphology information collected from an EM connectomic data set

may eventually be as powerful for the characterization of neuron types in a brain area as single-cell gene expression data, while additionally containing full connectivity information.'

The manuscript doesn't address if the point cloud representation is better than the previous views approach, therefore the novelty would be constrained to the performance of the method, not accuracy of output.

We indeed compared in-depth the accuracy of the point cloud and the multi-view representations (Fig. 1c,d and Supp. Text 1,2) which showed a similar level of prediction performance. We further observed slightly higher accuracy in at least one point cloud experiment (Supp. Text 1), albeit with additional myelin information. These results appear to be in line with recent findings by Goyal et al., 2021 (<https://arxiv.org/abs/2106.05304>). Importantly, the point cloud representation eliminates many pre- and post processing steps, which is very relevant for the massive data pipelines in connectomics.

As for other novel aspects of this toolkit, it is difficult to gather what updates have been implemented, regarding the authors' prior publications about the toolkit and method (references 11 & 18) and to other tools cited for extracting and analyzing synaptic connectivity.

We agree with this point, given the many improvements of SyConn2 over the original version and the short publication format. We therefore added a new Supplemental Table (Supp. Table 2) that makes the updates and features more explicit and enables easier comparison with other tools.

Data and Methods: although the methods are described extensively, as mentioned above, it is often unclear what particular improvements or differences were implemented with respect to referenced works. If the performance of the point cloud representation is the major aspect of novelty and significance of this toolkit, this may be better communicated by replacing either figure 1c or d with supplementary figure 1a. Regarding the significance of the methods, 2 examples are shown (figure 1e and f). Comments on the cell typing are given above.

We hope to have clarified the SyConn2 improvements by adding Supp. Table 2. The figure panels 1c or 1d add more value in our opinion, since they also show how much spatial context is required for accurate compartment and cell type classification with point clouds. Next to the faster processing (Supp. Fig. 1), we would like to stress that unsupervised cell type discovery with learned point cloud representations is an entirely novel approach (Fig. 1e).

Figure 1f, together with supplementary figure 2 give an illustrative exploration of inter-type differences for the distribution of mitochondria to synapses. Adding the synaptic size distributions for both cell types would help the reader to judge whether the difference shown might simply be an effect of synaptic size rather than mitochondria to synapse distance.

The figure caption already contains median values for the synaptic size distributions which are indeed different, a result expected from our side. We now additionally provide Supp. Fig. 2b which shows the full size distributions. However, we would like to stress that we explicitly compare small and large synapses

of the same cell type to each other in Figure 1f, with the purpose to control for the different cell-type specific synapse size distributions.

The evidence to show that this method can enable accurate cell typing and could be applied more widely is however not very strong from the one example provided (figure 1e).

We would like to refer to our response above.

It is not clear from which point cloud sampling-based embeddings are learned to allow semi-supervised cell typing.

The point cloud based embeddings were created as explained in the methods and in the caption for Fig. 1e. We think the approach is best described as unsupervised (since no manual labels were used) or self-supervised (since the training of the neural network was based on similarity assumptions of the underlying data, see Methods). Since this became not sufficiently clear, we have adapted the respective methods section.

Regarding data availability, a Neuroglancer-based browser allows access to neuron reconstructions labelled by type and subcellular compartments, including synapse junctions. Exploring the 2D and 3D data in the Neuroglancer framework is intuitive and already familiar to many users, thus a good choice for sharing the data more widely.

We agree and hope that the neuroglancer ecosystem will keep growing through collaborative efforts.

Conclusions and suggested improvements: the review above already suggests a number of major improvements.

Minor points:

- Citation styles are mixed, and not in a way consistent with inline versus note citations. For example, line 284 cites author inline, but others in same paragraph do not.

Done.

- Some section references do not correspond to section titles. For example, multiple references are made to “synapse extraction”, but actual methods section title is “synapse-cell association”.

Thanks, done.

- acronyms are often not written in full when first shown in the text. For example MSN and GP (line 83). Many acronyms are also not written in full in the figure legend.

Done.

- Figure 1: inconsistent cell type coloring between panels e, f

We have adjusted the colors to make them more consistent.

- Figure 1b: en passant not “en passent”

Done.

- line 130: “equally provided” should likely be “also provided” or simply “provided”

Done.

- line 145: "(binary)input" missing a space

Done.

- line 168: As written, this citation does not need to be inline/long-form.

Added zenodo reference.

- line 195: What is this RF trained on? The same ground truth for junction detection?

Thanks for pointing this out, this was missing in the methods and we have added it.

- line 340: "Neurons(-fragments)" missing a space

Done.

Clarity and context: one point regarding clarity: in line 97, it is stated that simple analyses can be done directly on the SyConn web-browser. It is not clear what type of analyses are intended, as this is a visualisation and exploration tool. Adding a brief example would be helpful.

We addressed this by pointing out in the main text and on the website more prominently that users can for example perform visual depth-first-search connectome queries to directly show the neurons of the strongest connectivity path downstream and upstream of a selected neuron. Such analyses, directly implemented in neuroglancer, are novel to our knowledge and might inspire the neuroglancer implementations of other connectome dataset providers.

Reviewer #2

Remarks to the Author:

Key results

The paper propose SyConn2, a cost-effective automated pipeline that can generate annotated connectome from input image volumes and do morphological analysis of the structures. Building upon previous work on automatic neuron reconstruction [1], the paper extend the work to reconstruct neurons with its ultrastructures (synaptic junctions, vessicles and mitchondria) and part segmentations (spine head, spine neck, axon, and etc.). Point cloud representation is used when doing part segmentation of neurons as well as cell classification (both under supervised and unsupervised setting). Finally, the reconstructed neurons are used to perform an analysis of the minimal mito-synapse distances.

From this, two key contributions can be identified.

1. Reconstruction of neurons with its ultrastructures and parts.
2. Using point cloud representation for part segmentation and cell classification.

Significance

This can greatly aid the research in connectome analysis. Even though there is previous work like [1,2] that perform segmentation of different structures, this work extract and combines different parts in a neuron as a single object (SuperSegmentationObject). This greatly improves the type of analysis that could be done.

We would like to thank the reviewer for recognizing how important it is to have an organized data representation to enable analyses. Connectomics still suffers from a lack of data standards, which will have to change over the next decade when the field will mature further, just as it has happened in genomics.

Data and methodology

Image data used are standard EM data used for connectome analysis.

The proposed pipeline is based on established deep learning approaches for analysing image volumes and point clouds.

Main claims in the paper are supported by quantitative analysis. Results on cell type classification and cell part segmentation (subcellular compartment models) is presented in supplementary materials (Supp. Text 1, 2). Cost analysis is presented in the paper (line 399). Some important experiments related to hyper-parameter tuning is also presented (paragraph starting from line 258).

To capture the uncertainty in the model, in Supp. Text 1. 10 fold cross validation is used and mean/std of the metrics are reported. In Supp. Text 2. average over 10 model checkpoints is reported.

Qualitative results can be found at <https://syconn.esc.mpcdf.mpg.de/>

Overall, we can observe very good neuron reconstructions. But it has to be noted that we can also see a noticeable amount of split errors (small false-positive regions inside/next-to neurons).

Conclusion

Qualitative results (<https://syconn.esc.mpcdf.mpg.de/>) present the best evidence towards the robustness, validity and reliability of the approach since its evaluated on the whole volume. Even though we observe some errors in the segmentations, overall pipeline is capable of successfully generating annotated connectome.

Suggested improvements / Unclear statements

Here I list several points (in order of their appearance in the text)

Line 29:

The term 'SyConn2' is introduced in line 29. Then in line 34, 39, 42 the term 'SyConn' is used and is referring to 'SyConn' proposed in [2]. But this is not explicitly introduced in the paper. Therefore, it would be better if 'SyConn' is also defined clearly.

We would like to thank the reviewer for pointing this out, the text was updated accordingly.

Paragraph starting at line 133:

Statement 1: "A myelin segmentation map (4-fold downsampled) was generated using SyConn's neural network model chunk inference pipeline..."

Statement 2: "For the myelin inference, a model based on the UNet architecture was used, which had the following parameters..."

Is the U-Net part of SyConn? If SyConn can produce myelin segmentation maps (according to statement 1), why a U-Net is necessary?

The U-Net is the architecture that was trained to perform the prediction with the distributed chunk inference pipeline in SyConn2. SyConn2 uses the ElektroNN neural network toolkit developed by us for connectomics applications (<https://github.com/ELEKTRONN>). ElektroNN features different model architectures, and we select and optimize the architectures for the different segmentation tasks.

Line 149:

For generating seeds, morphologically modified input segmentation maps has been used. Are these modified maps used as the final segmentation as well? If yes, why can we see small false negative regions (holes) in segmentation maps, for instance we can observe them inside the green and purple neurons in Fig 1.h. If not, why are they not used?

The modified maps are used only for the mitochondria and vesicle cloud instance segmentations. The cell instance segmentation maps were generated using an entirely different neural network model (flood-filling-neural networks, FFN), that produce instance segmentations directly. While binary morphological operations could also be beneficial in some cases for the cell segmentation (as in the case of Fig. 1h), neurons sometimes also get close to themselves (e.g., a spine almost touching the base dendrite), making such post-processing steps not as predictable as it may seem.

Line 221:

In this paragraph, use of point cloud representation is introduced. Why was point cloud representation selected over mesh representation? Mesh representation can better capture the shape and there are deep learning architectures that can process meshes similar to point clouds. Therefore, it might have produced competing or superior performance compared to the point cloud representation.

We agree that mesh representations could have been in principle an alternative for the SyConn2 toolkit (e.g., MeshNet, MeshCNN, MeshNet++). However, this would have introduced additional pre-processing effort, since currently the meshes for the cells are not extracted as a single connected component. This is a problem not to be underestimated, since individual cells can span very large volumes. We hope that future work by us or others will compare the performance of mesh-based vs point cloud-based approaches on connectomic data.

Paragraph starting at line 351:

How is the feature embedding for the unsupervised classification of cells computed? Based on this paragraph, the same network trained during supervised classification is used here. If that's the case, it's better to use the term 'pre-trained network' instead of 'unsupervised'. If it's not the case, the methodology used to train the network that produce the embedding is not clear.

It is the same architecture and the same input representation (point clouds), but with a different loss/target and both, the supervised and self-supervised/unsupervised model was trained from scratch. The supervised (with training labels) and self-supervised/unsupervised (triplet loss) training paradigms were used independently; we adapted the methods to make this more clear.

Line 361:

In this paragraph, an example of an analysis that could be performed with the SyConn2 connectome results is presented. It would be interesting, if authors can discuss such other avenues the connectome data produced by SyConn2 could be used (no experiments required).

Given the short article format, we have added two potential future research directions to the main text (plasticity rule analysis and neuromodulation), where we think that EM connectomics could make a contribution. We are happy to extend this discussion further if the editors give us the space.

References in the paper

Paper cites recent publications.

Minor suggestion:

Line 137: Citing 3D U-Net [3] is more accurate.

Done.

Reviewer's expertise

My expertise is in machine learning/computer vision.

References

1. Januszewski, M. et al. High-precision automated reconstruction of neurons with flood-filling networks. Nat. Methods 15, 605–610 (2018).
2. Dorkenwald, S. et al. Automated synaptic connectivity inference for volume electron microscopy. Nat. Methods 14, 435–442 (2017).
3. Çiçek O, et al. 3D U-Net: Learning Dense Volumetric Segmentation from Sparse Annotation. MICCAI (2016).

Decision Letter, first revision:

Dear Joergen,

Thank you for submitting your revised manuscript "SyConn2: Dense synaptic connectivity inference for volume EM" (N METH-BC47755A). It has now been seen by the original referees and their comments are below. The reviewers find that the paper has improved in revision, and therefore we'll be happy in principle to publish it in Nature Methods, pending minor revisions to satisfy the referees' final requests and to comply with our editorial and formatting guidelines.

TRANSPARENT PEER REVIEW

Thank you again for your interest in Nature Methods Please do not hesitate to contact me if you have any questions.

Best regards,
Nina

Nina Vogt, PhD
Senior Editor
Nature Methods

ORCID

Reviewer #1 (Remarks to the Author):

The authors have addressed the issues raised satisfactorily, implementing a number of changes primarily in the Methods and supplementary material. Table S1 is very welcome, very clearly stating the improvements to SyConn2, as compared to the previous iteration of the method. Regarding the cell typing, the distinction of additional cell types in figure 1e is useful, though some of these types do not form distinct clusters to one another. As the authors mention, it remains to be seen how useful this method will be for that particular purpose.

Minor comments:

Figure 1e:

a) can the authors please add the acronyms of the cell types to the figure legend? Only TAN is listed, with the remaining mentioned only in the Methods. That makes it hard to follow what is being referred to in this sentence “This analysis revealed additionally that Area X might harbor more cell types, for example local neurons that form synapses with excitatory ultrastructural characteristics---a neuron type in Area X that has so far only been physiologically identified²⁸ but not anatomically characterized” (page 2).

b) The colours chosen for FS and TAN are very similar (the latter a bit lighter), making it hard to assess possible overlap between data points or clusters. Could one of those be changed please?

Reviewer #2 (Remarks to the Author):

Authors have provided satisfactory responses to all the questions that were raised. I have added one detailed comment for the response corresponding to line 221. You can find all the responses in the attached file in blue text (section: Reviewer #2):

If you select an architecture like Voxel2Mesh, you will not need mesh ground truth to train the network. It could be trained with voxel ground truth (and it extract point clouds from it to train the mesh branch).

Author Rebuttal, first revision:

Response to referees

Second review round:

Reviewer #1 (Remarks to the Author):

The authors have addressed the issues raised satisfactorily, implementing a number of changes primarily in the Methods and supplementary material. Table S1 is very welcome, very clearly stating the improvements to SyConn2, as compared to the previous iteration of the method. Regarding the cell typing, the distinction of additional cell types in figure 1e is useful, though some of these types do not form distinct clusters to one another. As the authors mention, it remains to be seen how useful this method will be for that particular purpose.

Minor comments:

Figure 1e:

a) can the authors please add the acronyms of the cell types to the figure legend? Only TAN is listed, with the remaining mentioned only in the Methods. That makes it hard to follow what is being referred to in this sentence "This analysis revealed additionally that Area X might harbor more cell types, for example local neurons that form synapses with excitatory ultrastructural characteristics---a neuron type in Area X that has so far only been physiologically identified²⁸ but not anatomically characterized" (page 2).

Thanks for the useful suggestion -- we have added all cell type acronyms to the figure legend and additionally make it more explicit in the text which cell type we were referring to.

b) The colours chosen for FS and TAN are very similar (the latter a bit lighter), making it hard to assess possible overlap between data points or clusters. Could one of those be changed please?

Done.

Reviewer #2 (Remarks to the Author):

Authors have provided satisfactory responses to all the questions that were raised. I have added one detailed comment for the response corresponding to line 221. You can find all the responses in the attached file in blue text (section: Reviewer #2)

Thanks for the comment and pointing us to the reference!

Final Decision Letter:

Dear Joergen,

I am pleased to inform you that your Brief Communication, "SyConn2: Dense synaptic connectivity inference for volume EM", has now been accepted for publication in Nature Methods. Your paper is tentatively scheduled for publication in our October print issue, and will be published online prior to that. The received and accepted dates will be November 30th, 2021 and August 24th, 2022. This note is intended to let you know what to expect from us over the next month or so, and to let you know where to address any further questions.

Your paper will now be copyedited to ensure that it conforms to Nature Methods style. Once proofs are generated, they will be sent to you electronically and you will be asked to send a corrected version within 24 hours. It is extremely important that you let us know now whether you will be difficult to contact over the next month. If this is the case, we ask that you send us the contact information (email, phone and fax) of someone who will be able to check the proofs and deal with any last-minute problems.

If, when you receive your proof, you cannot meet the deadline, please inform us at rjsproduction@springernature.com immediately.

Once your manuscript is typeset and you have completed the appropriate grant of rights, you will receive a link to your electronic proof via email with a request to make any corrections within 48 hours. If, when you receive your proof, you cannot meet this deadline, please inform us at rjsproduction@springernature.com immediately.

Once your paper has been scheduled for online publication, the Nature press office will be in touch to confirm the details.

Content is published online weekly on Mondays and Thursdays, and the embargo is set at 16:00 London time (GMT)/11:00 am US Eastern time (EST) on the day of publication. If you need to know the exact publication date or when the news embargo will be lifted, please contact our press office after you have submitted your proof corrections. Now is the time to inform your Public Relations or Press Office about your paper, as they might be interested in promoting its publication. This will allow them time to prepare an accurate and satisfactory press release. Include your manuscript tracking number NMETH-BC47755B and the name of the journal, which they will need when they contact our office.

About one week before your paper is published online, we shall be distributing a press release to news organizations worldwide, which may include details of your work. We are happy for your institution or funding agency to prepare its own press release, but it must mention the embargo date and Nature Methods. Our Press Office will contact you closer to the time of publication, but if you or your Press Office have any inquiries in the meantime, please contact press@nature.com.

Please note that Nature Methods is a Transformative Journal (TJ). Authors may publish their research with us through the traditional subscription access route or make their paper immediately open access through payment of an article-processing charge (APC). Authors will not be required to make a final decision about access to their article until it has been accepted. Find out more about Transformative Journals

Authors may need to take specific actions to achieve compliance with funder and institutional open access mandates. If your research is supported by a funder that requires immediate open access (e.g. according to Plan S principles) then you should select the gold OA route, and we will direct you to the compliant route where possible. For authors selecting the subscription publication route, the journal's standard licensing terms will need to be accepted, including self-archiving policies. Those licensing terms will supersede any other terms that the author or any third party may assert apply to any version of the manuscript.

To assist our authors in disseminating their research to the broader community, our SharedIt initiative provides you with a unique shareable link that will allow anyone (with or without a subscription) to read the published article. Recipients of the link with a subscription will also be able to download and print the PDF. As soon as your article is published, you will receive an automated email with your shareable link.

Please note that you and your coauthors may order reprints and single copies of the issue containing your article through Springer Nature Limited's reprint website, which is located at <http://www.nature.com/reprints/author-reprints.html>. If there are any questions about reprints please send an email to author-reprints@nature.com and someone will assist you.

Best regards,
Nina

Nina Vogt, PhD
Senior Editor
Nature Methods